# Prediction of Soil Organic Carbon based on Landsat 8 Monthly NDVI Data for the Jianghan Plain in Hubei Province, China

**Yangchengsi Zhang [1], Long Guo [1,\*], Yiyun Chen [2], Tiezhu Shi [3], Mei Luo [1], QingLan Ju [1], Haitao Zhang [1] and Shanqin Wang [1]**

[1]  College of Resources and Environment, Huazhong Agricultural University, Wuhan 430070, China
[2]  School of Resource and Environmental Science, Wuhan University, Wuhan 430079, China
[3]  Key Laboratory for Geo-Environmental Monitoring of Coastal Zone of National Administration of Surveying, Mapping and GeoInformation & Shenzhen Key Laboratory of Spatial Smart Sensing and Services& College of Life Sciences and Oceanography, Shenzhen University, Shenzhen 518060, China
\*  Correspondence: guolong@mail.hzau.edu.cn; Tel.: +86-1363-864-5920

**Abstract:** High-precision maps of soil organic carbon (SOC) are beneficial for managing soil fertility and understanding the global carbon cycle. Digital soil mapping plays an important role in efficiently obtaining the spatial distribution of SOC, which contributes to precision agriculture. However, traditional soil-forming factors (i.e., terrain or climatic factors) have weak variability in low-relief areas, such as plains, and cannot reflect the spatial variation of soil attributes. Meanwhile, vegetation cover hinders the acquisition of the direct information of farmland soil. Thus, useful environmental variables should be utilized for SOC prediction and the digital mapping of such areas. SOC has an important effect on crop growth status, and remote sensing data can record the apparent spectral characteristics of crops. The normalized difference vegetation index (NDVI) is an important index reflecting crop growth and biomass. This study used NDVI time series data rather than traditional soil-forming factors to map SOC. Honghu City, located in the middle of the Jianghan Plain, was selected as the study region, and the NDVI time series data extracted from Landsat 8 were used as the auxiliary variables. SOC maps were estimated through stepwise linear regression (SLR), partial least squares regression (PLSR), support vector machine (SVM), and artificial neural network (ANN). Ordinary kriging (OK) was used as the reference model, while root mean square error of prediction ($RMSE_P$) and coefficient of determination of prediction ($R^2_P$) were used to evaluate the model performance. Results showed that SOC had a significant positive correlation in July and August (0.17, 0.29) and a significant negative correlation in January, April, and December (−0.23, −0.27, and −0.23) with NDVI time series data. The best model for SOC prediction was generated by ANN, with the lowest $RMSE_P$ of 3.718 and highest $R^2_P$ of 0.391, followed by SVM ($RMSE_P$ = 3.753, $R^2_P$ = 0.361) and PLSR ($RMSE_P$ = 4.087, $R^2_P$ = 0.283). The SLR model was the worst model, with the lowest $R^2_P$ of 0.281 and highest $RMSE_P$ of 3.930. ANN and SVM were better than OK ($RMSE_P$ = 3.727, $R^2_P$ = 0.372), whereas PLSR and SLR were worse than OK. Moreover, the prediction results using single-data NDVI or short time series NDVI showed low accuracy. The effect of the terrain factor on SOC prediction represented unsatisfactory results. All these results indicated that the NDVI time series data can be used for SOC mapping in plain areas and that the ANN model can maximally extract additional associated information between NDVI time series data and SOC. This study presented an effective method to overcome the selection of auxiliary variables for digital soil mapping in plain areas when the soil was covered with vegetation. This finding indicated that the time series characteristics of NDVI were conducive for predicting SOC in plains.

**Keywords:** NDVI time series; soil organic carbon; digital soil mapping; prediction model; machine learning methods

## 1. Introduction

Soil organic carbon (SOC), which is an essential nutrient of crop growth and the main carbon source and sink of greenhouse gases, influences agricultural production and global climate change [1–5]. The identification of the spatial distribution characteristics of SOC contributes to the investigation of the role of SOC in precision agriculture and the carbon cycle of the ecosystem. Digital soil mapping with the aid of easily obtained soil-forming factors, such as terrain, climatic, and vegetation factors, can continuously map the spatial distribution of SOC [6–9]. However, the spatial characteristic of natural landscape is similar in plains or flat terrain areas, and most traditional soil-forming factors exhibit small spatial variations that prevent them from contributing to the development of soil–landscape models. Thus, the selection of suitable auxiliary variables to complete soil mapping in plain areas is challenging.

The traditional measurement of SOC content is based on the laboratory analysis of field soil sampling [10–12]. The soil data of sampling points are discrete and incapable of providing continuous and complete information regarding the total study area and require extensive time and labor. The spatial variability of SOC through field soil sampling cannot be obtained. Numerous studies on the digital mapping of SOC have been conducted to resolve this issue on the basis of the spatial autocorrelation of soil [13–16]. Many studies have proven that soil properties exhibit strong spatial dependence between neighboring regions, and trend surface analysis, inverse distance weighted, and geostatistical models have been successfully used in soil mapping [17–19]. However, such methods merely rely on the correlation among soil sample points, which is limited by the geographical location of sampling points. In other words, traditional geostatistical methods based on geospatial autocorrelation have two limitations, namely, they are locally limited by sampling density [20,21] and ignore the role of environmental factors, thereby causing the results to be inconsistent with reality [22,23]. These methods encounter difficulty in describing the spatial distribution characteristics of SOC in complex terrains.

In general, the occurrence, formation, and degradation of soil are influenced by the interaction of the surrounding environmental factors for a long period of time. Thus, numerous soil-forming factors have been used to develop soil–landscape models. Wang et al. [4] estimated the SOC distribution by using nine environmental variables (e.g., precipitation, temperature, land use, and elevation) with boosted regression trees. Song et al. [24] mapped the SOC content through geographically weighted regression using several environmental predictors (e.g., slope, aspect, elevation, land use, and normalized difference vegetation index [NDVI]) in a case study of Heihe River Basin, China. Wang et al. [25] estimated the SOC spatial distribution using a weighted regression approach based on the correlation of environmental variables (NDVI, annual precipitation and average temperature and moisture index). Thus, these demonstrate that large variation in topography creates large variations in climate and other environmental variables related to SOC, leading to strong statistical relationships. However, environmental variations in areas with small topography, such as plains, are small, making the development of the accurate predictions of SOC difficult [26–28]. The variation of soil properties is a comprehensive result of the long-term interaction of various environmental factors. Thus, responding to the spatial heterogeneity of soil properties through environmental factors with small differences is difficult, especially in small-scale areas where the variation of environmental variables is obscured. Hence, suitable environmental variables should be selected to determine the spatial variation characteristics of SOC in flat areas and utilize them for SOC mapping and precision agriculture.

Several scholars have identified many other alternative factors to respond to the spatial variation of soil properties and solve the difficulty in selecting environmental factors in plain areas or flat terrain regions. Zhu et al. [29] and Liu et al. [30] presented a new land surface dynamic feedback (LSDF)

model by comparing the temporal responses to a rainfall event to map soil texture, SOC, and other properties. The LSDF model combined with land surface spectral or temperature variations uses short-time remote sensing images to predict soil properties [31]. The contradiction between return time and spatial resolution limits the development of high-precision soil maps. Hyperspectral images have been used to quantitatively predict soil properties through the spectral reflectance of the surface soil [32,33]. However, surface vegetation and scant hyperspectral images hinder their use in large areas.

Agricultural land occupies approximately 38.18% of the world area based on the data of World Bank in 2016 and the main land use type among all land use types. Agricultural production and activity constantly influence the change of SOC storage [34,35]. Thus, the spatial and temporal distribution rules of SOC in agricultural lands should be investigated. SOC as the main soil fertilizer influences the soil structure and crop growth [36]. With the improvement of remote sensing technology, increasing studies have focused on recording the growth status of crops in different phenological periods [37–39]. VI is designed to enhance the contribution of vegetation properties and allow the reliable spatial and temporal intercomparisons of terrestrial photosynthetic activity and canopy structural variations [40]. Wang et al. [41] concluded that an obvious exponential relationship exists between broadband NDVI and gross primary productivity (GPP), while a linear relationship occurs between broadband NDVI and the fraction of absorbed photosynthetically active radiation (fAPAR). This condition indicated that the VIs of remote sensing images can record crop variation. Thus, from this perspective, remote sensing VIs may be used to reflect the spatial variation of soil properties (e.g., SOC). However, at present, many studies have only considered environmental variables (e.g., NDVI time series characteristics) at a certain time when modeling with environmental elements and ignored the variability with time. Kheir et al. [42] used NDVI data from April 1987 as the parameter in modeling. Burnham and Sletten [43] adopted an NDVI map from a July 26, 2004 image when mapping the spatial distribution of SOC. Taghizadeh-Mehrjardi et al. [44] mapped SOC using data mining techniques with some ancillary data that included ratio VI (RVI), soil-adjusted VI (SAVI), and NDVI on March 28, 2013. However, the temporal characteristics of environmental variables should be considered because the variation of soil physical and chemical properties requires time.

In previous studies, NDVI was used as an important index to monitor the growth status and cover the vegetation degree. NDVI played an important role in remote sensing applications. However, considering only NDVI at a certain point in time may cause unrealistic results because the sequential feature of NDVI data can reveal additional information about the study object for a time and lead a highly comprehensive approach. Numerous studies based on VI time series have been applied in various fields. Shen et al. [45] extracted winter wheat information on the basis of time series NDVI in the Guanzhong area. Li et al. [46] analyzed the land damage and recovery process in a rare earth mining area using multisource sequential NDVI. Wardlow and Egbert [47] evaluated the applicability of time series MODIS 250 m NDVI data for large-area crop-related LULC (land use/land cover mapping on the U.S. Central Great Plains. Testa et al. [48] estimated the phenological metrics in French deciduous forests using MODIS-derived EVI, NDVI, and WDRVI time series. Nagy et al. [49] used MODIS NDVI time series to forecast wheat and maize yields on the Tisza River catchment and reported crop statistics. These methods show the valuable role of NDVI time series data in qualitative analysis. In this study, NDVI time series is closely related to crop yield (related to the sum of vegetation gross primary production). Ichii et al. [50] used satellite-based time series observations (including NDVI time series) and four process-based terrestrial biosphere models to identify and understand the changes of terrestrial GPP in Asia and obtained credible results. Burnham and Sletten [43] observed a remarkable relationship between NDVI and SOC storage. Wang et al. [4] concluded that NDVI is highly predictive of SOC contents that reflect vegetation productivity and biomass. These findings indicate that a strong correlation exists between NDVI and SOC, which may be deeply connected in plain areas. Thus, the current research attempted to extract the valuable information of NDVI time series data in the digital soil mapping of SOC and determine the optimal model for predicting SOC in a plain region on the basis of previous studies. This study represented a new concept that provides

a convenient predictive method to capture unavailable soil property information in a plain region, especially when the soil on earth surface is covered by vegetation.

Jianghan Plain is one of the important food production regions in China. The topography of Jianghan Plain is dominated by plains, and its topographical relief is uniform. Honghu City, which is located in the middle of Jianghan Plain, was selected as the study region. NDVI maps from January to December were used as the auxiliary variables, and five methods, namely, stepwise linear regression (SLR), ordinary kriging (OK), partial least squares regression (PLSR), support vector machine (SVM), and artificial neural network (ANN), were applied as the predictive methods. The objectives of this study were to (1) discuss the correlation between SOC and NDVI time series data, (2) explore the feasibility of using NDVI time series data in SOC mapping, and (3) compare the differences of the digital mapping of SOC through different predictive methods.

## 2. Materials and Methods

### 2.1. Study Area and Sampling

Honghu City (113°07′–114°05′E, 113°07′–114°05′N), Hubei Province, which is located at the center of Jianghan Plain, was selected as the study area. Honghu has an approximate area of 2519 km$^2$ and an average altitude ranging mostly from 23 m to 28 m. The study area is an alluvial plain. Thus, the terrain has gentle slopes, with an average slope of approximately 0.3%. The main soil parent materials of this area are river alluvium and lacustrine deposits. The soils are diverse and composed of various Chinese soil taxonomy classifications, including paddy and moisture soils [22]. Their approximate classifications from the World Reference Base of Soil Resources are Typic Haplaquept and Dystrochrept [51]. The soil styles are simple, but the soil nutrient varies. The main land use can be classified as cultivated, forest, construction land, water, and other lands. The cultivated land area accounts for 66.49% of the entire region, with rice, oilseed rape, lotus root, and wheat as staple crops. Generally, the rice in Hubei Province matures two or three times a year, and oilseed rape matures once a year, which is sown in September or October, and harvest time is in April or May the following year. Honghu City experiences subtropical humid monsoon climate with four distinct seasons. Rainfall is abundant and varies between 1060.5 and 1331.1 mm. The mean annual temperature is approximately 16 °C, with the coldest temperature of 3.8 °C in January and hottest temperature of 28.9 °C in July.

A total of 787 samples (0–30 cm) were collected through random sampling on the main study area. The distances between each sampling site ranged from 100 m to 1492 m. The spatial distribution of these soil sampling points was mainly concentrated in the central and northwestern parts of Honghu City (Figure 1). For each location, five soil samples were taken from the four corners and the center of a 1 m × 1 m square area. Approximately l kg subsample was collected from the composite sample for laboratory analysis after plants and debris covering the soil surface were removed. First, all samples were air dried in the laboratory at 20 °C–25 °C for 14 days. Subsequently, these soils were crushed using a porcelain mortar to break down large aggregates and passed through a 0.25 mm sieve. This step can remove the effect of soil texture and moisture and reflect the authentic SOC characteristics of soil samples. Subsequently, the SOC content was measured using potassium dichromate [52]. All these methods can be found in the studies of Guo et al. [53] and Liu et al. [22].

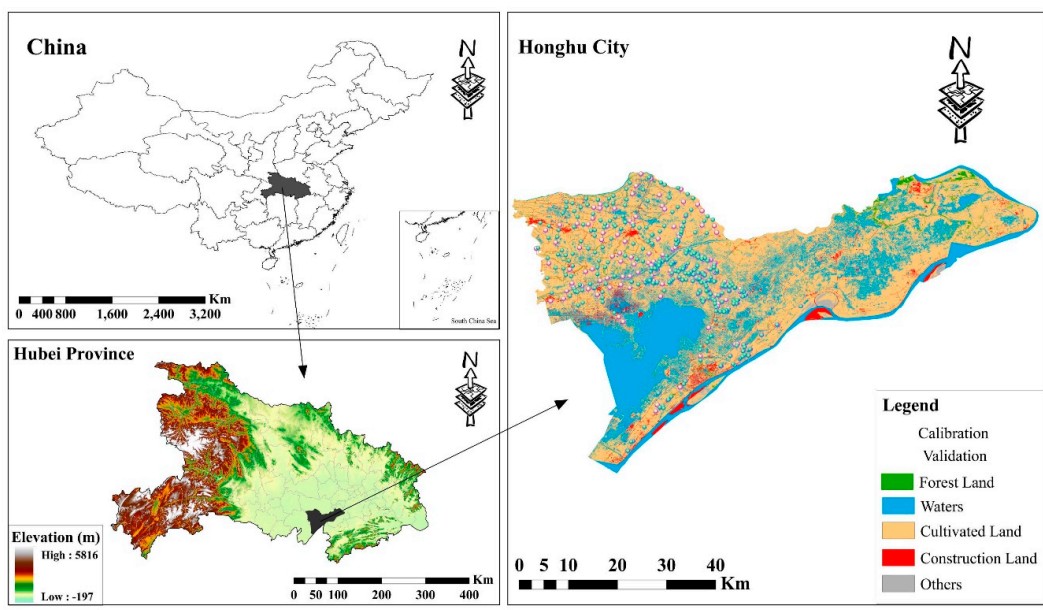

**Figure 1.** Location of the study area and spatial distribution of the calibration (407) and validation (271) datasets.

## 2.2. Data Source and Processing

Landsat 8 OLI imageries with 30 m spatial resolution and <10% cloud cover from January to December in 2013 were downloaded from the Geospatial Data Cloud. In particular, although the OLI 2013-12-06 image presented 49.93% cloud coverage, the study area was clear and without cloud cover for all image data. Other image data in the same month from adjacent years (2014–2016) were selected to replace certain months that exhibited insufficient image data or substantial cloud cover. Table 1 shows the classified parameters of these remote sensing images. Radiometric calibration and atmospheric correction through the tools of Radiometric Correction in ENVI 5.1 were used to handle the data. One remote image of resource satellite three-01 A with 2.1 m spatial resolution (November 28, 2013) was interpreted into five land-use types through supervised classification in the ENVI 5.1 tool (a method of SVM classification based on fuzzy recognition) with an overall accuracy of 90.27% and a kappa coefficient of 0.827. The land-use types were cultivated, forest, and construction land, waters, and other land (Figure 1). For Landsat 8 OLI, the best representations of vegetation growth and variable, visible red band 4 (630–680 nm) and near-infrared band 5 (845–885 nm), respectively, were used to construct an NDVI model. This index was calculated as follows:

$$\text{NDVI} = \frac{(\text{Band5} - \text{Band4})}{(\text{Band5} + \text{Band4})}, \tag{1}$$

**Table 1.** Main parameters of remote sensing images.

| Num. | Date | Path/Row | Cloud Cover (%) | ID |
|---|---|---|---|---|
| 1 | 2014-01-23 | 123/39 | 19.72 | LC81230392014023LGN00 |
| 2 | 2016-03-01 | 123/39 | 3.87 | LC81230392016061LGN00 |
| 3 | 2015-03-31 | 123/39 | 8.99 | LC81230392015090LGN00 |
| 4 | 2013-04-26 | 123/39 | 2.25 | LC81230392013116LGN01 |
| 5 | 2013-05-28 | 123/39 | 1.49 | LC81230392013148LGN01 |
| 6 | 2013-06-13 | 123/39 | 0.28 | LC81230392013164LGN00 |
| 7 | 2013-07-31 | 123/39 | 1.22 | LC81230392013212LGN00 |
| 8 | 2013-08-16 | 123/39 | 13.88 | LC81230392013228LGN00 |
| 9 | 2013-09-17 | 123/39 | 0.12 | LC81230392013260LGN00 |
| 10 | 2015-10-25 | 123/39 | 20.03 | LC81230392015298LGN00 |
| 11 | 2015-11-26 | 123/39 | 6.70 | LC81230392015330LGN01 |
| 12 | 2013-12-06 | 123/39 | 49.93 | LC81230392013340LGN00 |

Many studies have shown that crop production is related to temperature and precipitation [54–56]. Figure 2 shows the small variation of temperature and precipitation in the study area from different years (2013–2016). Thus, the monthly data are close to the long-term average, and the data in Table 1 could be considered an artificial time series.

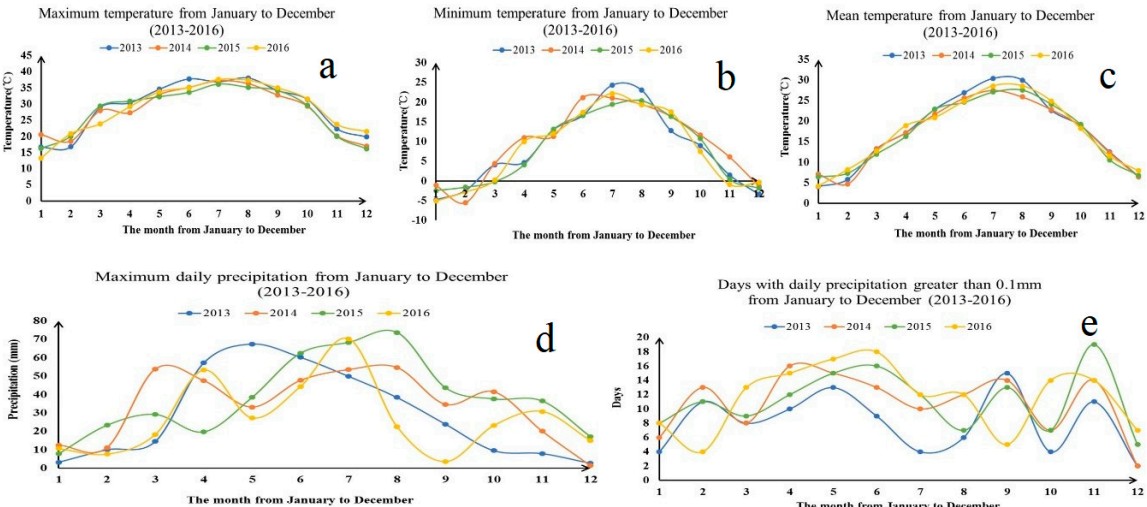

**Figure 2.** Change of climate conditions in Honghu City from January to December, 2013–2016, (**a**): the maximum temperature, (**b**): the minimum temperature, (**c**): the mean temperature, (**d**): the maximum daily precipitation and (**e**): days with daily precipitation greater than 0.1 mm.

### 2.3. Prediction Models

A total of 787 soil sampling points was extracted from the NDVI data using the Spatial Analyst tool box of ArcGIS 10.4. After the abnormal values were rejected by the rule of 3δ criteria (eliminate "outliers") [57], 678 sampling sites (not under cloud cover) were chosen from 787 soil samples, which completely belonged to cultivated land, to develop the SOC prediction models. The 678 soil samples were divided into the training dataset that comprises 407 (60%) samples and the validation dataset that consists of 271 (40%) samples.

### 2.3.1. Stepwise Linear Regression (SLR) Model

The SLR model was used to construct the relationships between SOC and time series NDVI data [44]. The SLR model can be expressed as follows:

$$Z(x_0) = \beta_0 + \sum_{i=1}^{n} \beta_i * p_i + \varepsilon, \tag{2}$$

where $Z(x_0)$ is the estimated variable of SOC, $\beta_0$ is a constant term, $\beta_i$ represents the regression coefficients, $p_i$ denotes the independent variables of time series NDVI data, and $\varepsilon$ indicates the residuals. The SLR model was operated on SPSS statistics 23.

### 2.3.2. Ordinary Kriging (OK) Model

OK is widely applied in the spatial prediction of soil properties. The core theory of OK is that spatial autocorrelation is based on the spatial autocorrelation of soil properties. The soil attributes of unsampled points can be predicted by giving weights to the surrounding observation points [17]. This technique is optimal because it provides unbiased estimates with minimum and known errors [22]. The predicted model can be expressed as follows:

$$Z^*(x_0) = \sum_{i=0}^{n} \lambda_i * Z(x_i), \tag{3}$$

where $Z^*(x_0)$ is the estimated SOC value of variable $Z$ at location $x_0$, $Z(x_i)$ is the measured SOC data, $\lambda_i$ represents the weights combined with the measured values, and n is the number of measured values within a neighborhood of four or eight.

In this study, the OK model was generated by using a spherical variogram model on ArcGIS 10.4. The equation of the spherical model can be defined as follows:

$$\gamma(h) = \begin{cases} 0 & h = 0 \\ C_0 + \left(\frac{3h}{2a} - \frac{h^3}{2a^3}\right) & 0 < h < a \\ C_0 + C & h > a \end{cases}, \tag{4}$$

where $h$ is the spatial lag between two locations, a is the range, $C_0$ is the nugget value, and $C_0 + C$ is the partial sill.

### 2.3.3. Partial Least Squares Regression (PLSR) Model

Compared with the abovementioned approaches, machine-learning methods can effectively solve the collinearity problems existing in environmental variables to ensure that the proposed soil prediction model can have an expected immense effect. Wold et al. [58] proposed a new multivariate statistical analysis method, namely, PLSR, which integrates the advantages of multiple linear regression, principal component, and typical correlation analyses to effectively solve the multicollinearity among environmental factors. PLSR is a popular modeling technique used in chemometric and quantitative spectral analyses. It is based on a linear transition from numerous original descriptors to a new variable space based on a small number of orthogonal factors [53,59]. This method is used to create predictive models when many highly collinear predictor variables exist [60]. Tahmasbian et al. [61] considered that PLSR determines a few linear combinations (latent variables (LVs)) of the original X-values and uses only those linear combinations in the regression equation. This method discards irrelevant, superfluous, and unstable information and uses the most relevant X-variation for regression analysis [62,63]. Thus, PLSR enables a soil prediction model to have strong stability and excellent predictive ability [64]. The tool package of PLS_Toolbox_811_installer in MATLAB 2017 was used to implement the PLSR. The detailed theory can be found in Wold et al. [58].

### 2.3.4. Support Vector Machine (SVM) Model

SVM for regression (SVR) is a kernel-based learning regression method from statistical learning theory, which was proposed by Cherkassky [65]. It is based on the computation of a linear regression function in a multidimensional feature space. The linear model created in the new space can represent a nonlinear decision boundary in the original space [44]. SVMs for regression aim to create an optimal hyperplane that can separate classes and create the widest margin between their data or fit data and predict with minimal empirical risks and modeling function complexity [66]. Figure 3 illustrates the operating principle of the SVM model. This method has been widely applied in regression and forecasting in various fields, such as agriculture, meteorology, and environmental monitoring studies [67–69].

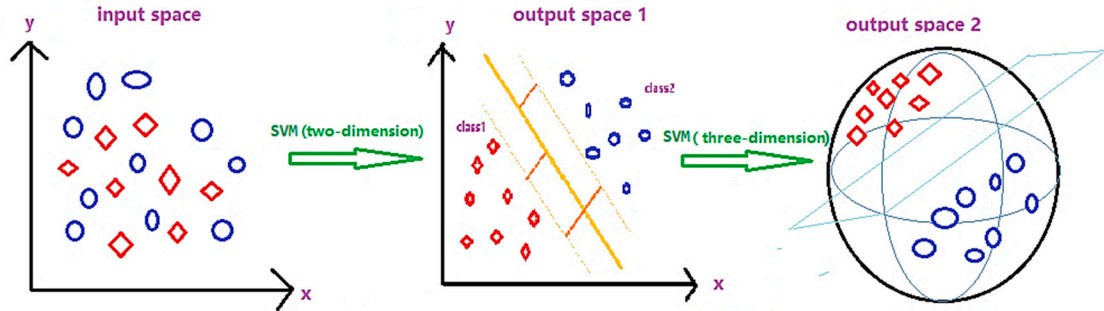

**Figure 3.** Support vector machine (SVM) schematic.

The SVM model exhibits several parameters. Cost (*c*) represents the penalty associated with errors larger than epsilon (*ε*). Increasing cost value causes a close fitting to the training data. The kernel gamma (*γ*) parameter controls the shape of the separating hyperplane. Similarly, increasing gamma usually increases the number of support vectors. For epsilon (*ε*), no penalty is associated with points predicted within the distance epsilon (*ε*) from the actual value in training the regression function. A decreasing epsilon forces close fitting to the training data. The nu (*v*) parameter indicates a lower bound on the number of support vectors to use, which is given as a fraction of total calibration samples, and an upper bound on the fraction of training samples, which are errors [70].

In addition, two versions of SVM regression, namely, "epsilon(*ε*)-SVR" and "nu(*v*)-SVR," are commonly used in the SVM model. The original SVM formulations for regression (SVR) use parameter cost (c) and epsilon (*ε*) to apply a penalty to the optimization for points that are incorrectly predicted. SVR is replaced by another parameter, nu (*v*), which applies a slightly different penalty. The main incentive for "nu (*v*)" version of SVM is its highly meaningful interpretation, which is attributed to "nu (*v*)" representing an upper bound on the fraction of training samples, which are errors (poorly predicted), and a lower bound on the fraction of samples, which are support vectors. Epsilon (*ε*) and nu (*v*) are the different versions of the penalty parameter. The same optimization problem is solved in either case [70].

2.3.5. Artificial Neural Network (ANN) Model

The ANN algorithm simulates the human learning processes by establishing and reinforcing the linkages between the input and output data [66]. It is based on the data processing in biological nervous systems because numerous cells exist for the reception of information, others for forwarding and storage, and another group for the outward release of information [60]. The A neural network (NN model is selected for several reasons. First, the approach has been successful in estimating the SOC content using ancillary data [44]. Second, ANN provides excellent modeling capabilities for complex, noisy environmental datasets where the relationship between input and output parameters is not well understood. Finally, NNs are frequently considered "black box" models because extracting the information they develop is easier on the modeled system than with other approaches [71]. Thus, the approach is convenient to use. Taghizadeh-Mehrjardi et al. [44] concluded that ANNs are the best method to predict the SOC content compared with different data mining methods and algorithms. Such modeling methods and techniques can potentially explore nonlinear relationships between various complicated variables and are thus proven to be highly powerful for digital SOC mapping. In the present study, NDVI time series are used as input parameters in the ANN model for prediction. The technical principles and processes can be found in Basheer and Hajmeer [72] and are shown in Figure 4.

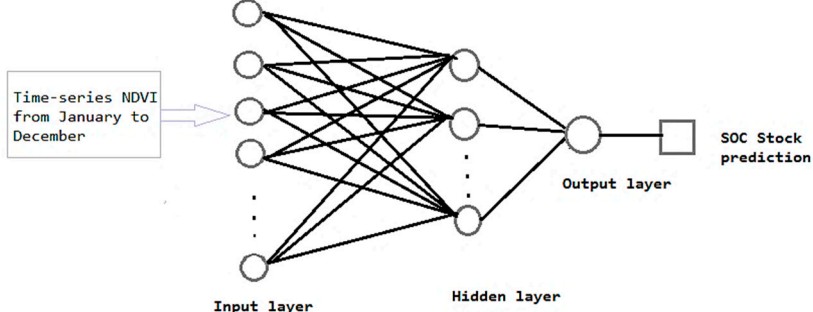

**Figure 4.** Artificial neural network (ANN) schematic.

A highly common type of ANN is the multilayer perceptron. The network contains interconnecting nodes called neurons that are connected to one another through weighted synapses. First, the random initial raw values of weights are placed into the synapses to make an ANN model. Thereafter, these data are increasingly corrected during training phase. Subsequently, the computed outputs of the network are contrasted with the actual values. Finally, the errors are backpropagated to adjust the weight values for minimizing errors [44,73].

### 2.4. Model Validation and Evaluation

The validation datasets were used to analyze the predictive performance of the models for comparing the predicted results and determining the optimal model. The accuracy assessment of these models was evaluated using the root mean square error (RMSE) and coefficient of determination ($R^2$) as the performance indicators. RMSE statistics were used to evaluate the consistency between the estimated values from the models and observed ones from the field (i.e., measurement of prediction accuracy). $R^2$ reflects the degree of fitting between the measured and predicted values.

These indexes were calculated as follows:

$$\text{RMSE} = \sqrt{\frac{1}{n} \sum_{i=1}^{n} (M_i - P_i)^2}, \tag{5}$$

$$R^2 = \frac{\sum_i^n \left(P_i - \overline{M_i}\right)^2}{\sum_i^n \left(M_i - \overline{M_i}\right)^2}, \tag{6}$$

where $M_i$ and $P_i$ are the measured and predicted SOC values at site $i$, respectively, n is the total amount of modeling data, 678, *Cov* indicates the covariance between the measured and predicted values, *Var* is the variance between the measured and predicted values, and $\overline{M_i}$ is the average value of measured SOC.

## 3. Results

### 3.1. Basic Statistics of SOC and NDVIs

The SOC content of the samples collected in this study ranged from 0.32 g kg$^{-1}$ to 33.95 g kg$^{-1}$ of the topsoil (0–30 cm) with a mean value of 6.24 g kg$^{-1}$. Table 2 shows that the mean values of the total calibration and validation datasets were 6.24, 6.72, and 6.00 g kg$^{-1}$, with standard deviations (SDs) of 5.02, 5.64, and 4.70, respectively. The basic statistics of calibration and validation datasets were similar to those of the entire dataset. This finding indicated that the calibration and validation datasets can effectively represent the entire dataset. The calibration dataset presented a wider range (33.51 g kg$^{-1}$) and larger minimum (0.44 g kg$^{-1}$) and maximum (33.95 g kg$^{-1}$) values compared with the validation dataset (25.26, 0.33, and 25.59 g kg$^{-1}$, respectively). Thus, the SOC values of the validation dataset were entirely included in the calibration dataset to ensure prediction authenticity. SD indicated that the

SOC values were generally stable (fluctuated at approximately 5.50), which were contrary to the results showed by the coefficient variation (CV). CV was adopted to illustrate the SOC variability. For soil properties, variability ranking was assigned on the basis of the classification proposed by Wilding [74], where CV < 15% is the least variable, 15% < CV < 35% is the moderate variable, and CV > 35% is the most variable. The CV totality, points in croplands, calibration, and validation of SOC were 80.44%, 82.27%, 83.93%, and 78.33%, respectively, which exhibited strong variability. The skewness and kurtosis of all datasets denoted that the SOC data had an approximately normal distribution. Thus, these data can be used to create models for SOC prediction in the study area.

**Table 2.** Basic statistics of SOC (soil organic carbon) (g kg$^{-1}$) in totality, cultivated land, calibration, validation.

| Item | Num | Min | Max | Mean | Range | SD | CV (%) | Skewness | Kurtosis |
|---|---|---|---|---|---|---|---|---|---|
| Totality | 787 | 0.32 | 33.95 | 6.24 | 33.63 | 5.02 | 80.44 | 2.25 | 5.42 |
| Points in croplands | 678 | 0.33 | 33.95 | 6.43 | 33.62 | 5.29 | 82.27 | 2.13 | 4.53 |
| Calibration dataset | 407 | 0.44 | 33.95 | 6.72 | 33.51 | 5.64 | 83.93 | 2.07 | 3.97 |
| Validation dataset | 271 | 0.33 | 25.59 | 6.00 | 25.26 | 4.70 | 78.33 | 2.14 | 5.17 |

Note: Min is minimum, Max is maximum, SD is standard deviation, and CV is coefficient variation.

The spatial variation of terrain factors in Honghu City is shown in Figure 5, and their Pearson's correlation coefficients (*r*) with SOC are represented. These maps exhibited that topographic factors had weak spatial variation, and most of the study area had similarly low values. For instance, DEM ranged from −86 m to 154 m with a low *r* of 0.129, indicating a weak correlation with SOC. Similarly, the same spatial characteristics appeared in the rest of terrain factors (slope, surface roughness, and topographic position index (TPI)), and their Pearson's correlation coefficients were 0.021, 0.014, and 0.044, respectively, showing frail relationship with SOC. Thus, the effect of terrain factors for SOC in this low-relief area was insignificant, and the spatial variation of SOC cannot be obtained by the terrain parameters.

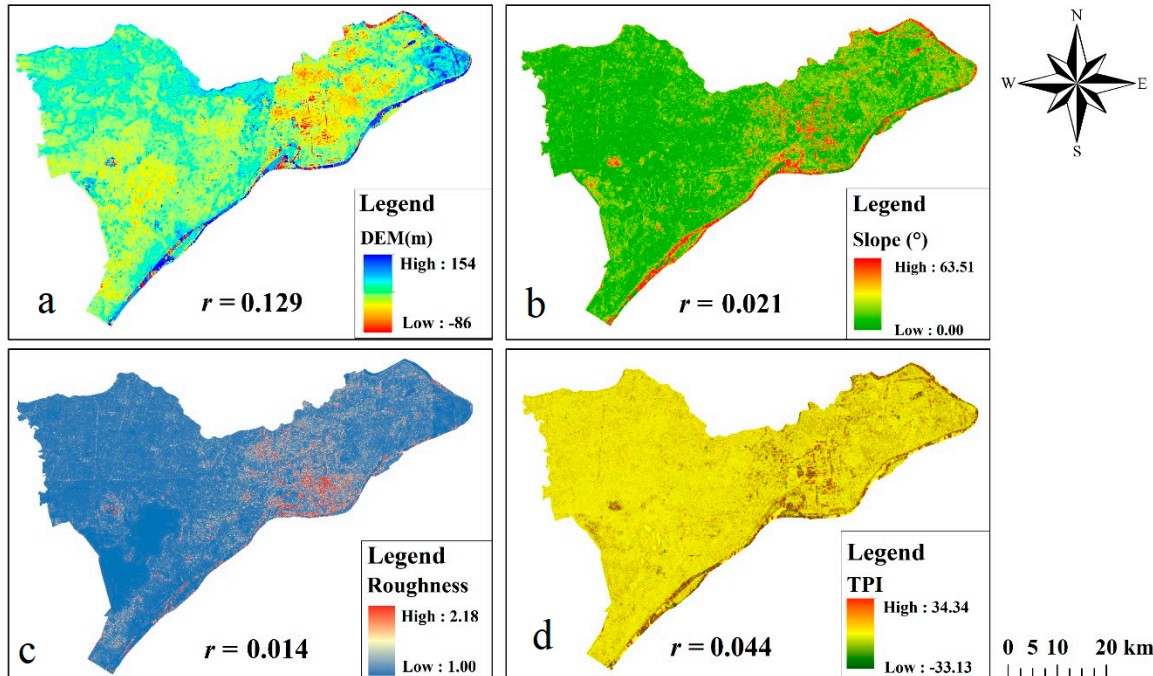

**Figure 5.** Terrain factors ((**a**): DEM, (**b**): slope, (**c**): roughness and (**d**): TPI) of the study area and their Pearson's correlation coefficients (*r*) with SOC (soil organic carbon). *r* is the Pearson's correlation coefficients between terrain factors, and SOC, TPI is the topographic position index.

The time series NDVI data of the soil samples from January to December in one year are appended as Figures 6 and 7. The highest NDVI appeared in July with the highest peak, followed by August, March, and November with lower peaks. The lowest NDVI was observed in May with the lowest valley, followed by October, December, and February. In the study region, agricultural land was the main land-use type, and surface vegetation landscape was influenced by agricultural cultivation. Thus, the spatial and temporal characteristics of NDVI were determined by the phenophase of crop growth. The main summer crops were rice, lotus root, and cotton, and the main winter crops were oilseed rape (*Brassica napus* L.) and wheat. Winter crops were sown in September or October and harvested April or May the following year. Summer crops were sown in April or March and harvested in September or October. Thus, NDVI showed high values during these months (e.g., August and March) when crops matured (Figure 7). However, NDVI values were relatively low in these periods because of the harvest of oilseed rape and wheat in April or May and rape and rice crops in September or October.

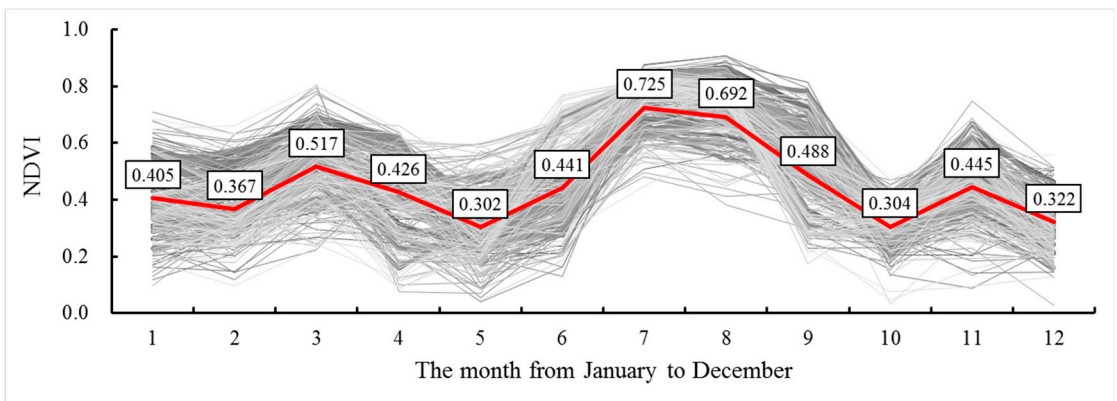

**Figure 6.** Variation of NDVI (normalized different vegetation index) of the soil samples from January to December.

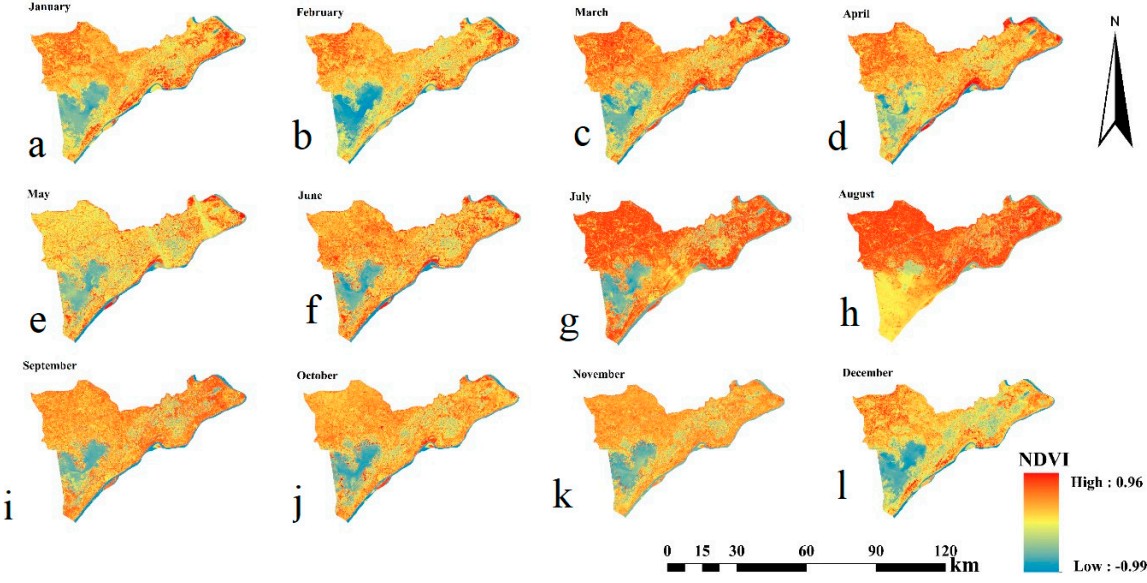

**Figure 7.** Spatial distribution of NDVIs (normalized different vegetation index) from January to December. The figures of (**a**–**l**) are corresponding the months of January to December.

### 3.2. Relationship Between SOC and NDVI Time Series Data

Figure 8 presents the correlation analysis results of the SOC content and NDVI by Pearson correlation coefficient (*r*) 0. A significant positive correlation is observed between the SOC content and NDVI in July and August. By contrast, a significant negative correlation exists between the SOC

content and NDVI in January, February, March, April, November, and December. This finding could be explained by the mature rice, cotton, and lotus root from July to August, thereby presenting a significant positive correlation. In January, February, March, April, November, and December, the negative correlation may be caused by the incomplete maturity of crops and exposure of soil on the surface. The harvest of oilseed rape in May and June and rice and lotus root in September and October leads to the earth surface covered with soil. Thus, the correlation between the SOC content and NDVI in May, June, September, and October is insignificant. These results demonstrate that NDVI time series data are closely related to crop growth cycle, which may be deeply affected by the SOC content under vegetation in these months and could be used to predict the spatial distribution of SOC.

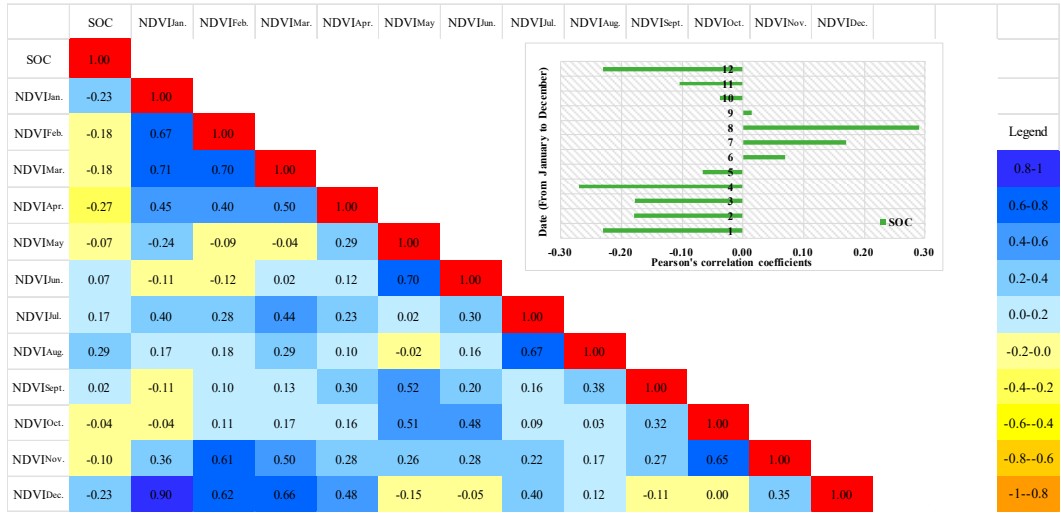

**Figure 8.** Pearson correlation coefficient between SOC (soil organic carbon) and NDVI (normalized difference vegetation index) from January to December. $NDVI_{Feb.}$ is the NDVI data in February, and $NDVI_{Jul.}$ is the NDVI data in July, etc.

### 3.3. Prediction of SOC Using Different Predictive Models

The NDVI time series data of 12 months were used as predictors in the SLR model. After variables that caused multicollinearity or nonsignificant were filtered and removed, the NDVI datasets in January, February, April, June, July, August, and September were used as the auxiliary variables in creating the SLR model. Table 3 shows the detailed parameters of SLR. The unstandardized coefficients showed the influence of NDVI variables on SOC with dimension, and the normalized coefficient showed the influence without dimension. All independent variables showed the influence degrees of NDVIs to SOC in different months because they had the same unit. The coefficients of NDVI showed that NDVIs in February, September, January, and April and July, August, and June had negative and positive relationships with SOC. The NDVI in September exhibited the highest coefficient (−11.22) among all months, whereas that in April presented the lowest coefficient (−4.78). Thus, these predictors explained most of the spatial variations of SOC, which showed relatively strong correlation with SOC. Conversely, the absolute values of beta in the rest of months were slightly lower than suggested in these months, with NDVI demonstrating a weak effect on SOC. Thus, the influences of NDVIs to SOC varied with months and revealed the relationships between vegetation and SOC. Meanwhile, the significance of all independent variables used in calculation was less than 0.05. This result was credible from the perspective of statistics.

**Table 3.** Stepwise multivariate regression model analysis.

|  | Unstandardized Coefficients | | Normalized Coefficient | t | Significance |
|---|---|---|---|---|---|
|  | Beta | Standard Deviation | Beta | | |
| (constant) | 7.42 | 1.44 | – | 5.15 | 0.00 |
| NDVI$_{Feb.}$ | −7.17 | 2.69 | −0.17 | −2.67 | 0.01 |
| NDVI$_{Jul.}$ | 8.09 | 2.86 | 0.20 | 2.84 | 0.01 |
| NDVI$_{Sept.}$ | −11.22 | 1.85 | −0.33 | −6.07 | 0.00 |
| NDVI$_{Jan.}$ | −5.12 | 2.50 | −0.14 | −2.05 | 0.04 |
| NDVI$_{Aug.}$ | 5.20 | 1.88 | 0.19 | 2.77 | 0.01 |
| NDVI$_{Jun.}$ | 5.30 | 1.81 | 0.15 | 2.93 | 0.00 |
| NDVI$_{Apr.}$ | −4.78 | 2.28 | −0.12 | −2.10 | 0.04 |

Notes: NDVI$_{Feb}$ is the NDVI (normalized different vegetation index) values in February, and NDVI$_{Jul}$ is the NDVI values in July, and so on.

Figure 9 shows the experimental variogram of the OK model. The best fit variogram model was a spherical one with a range of 308.97 m, nugget value of 21.67, and partial sill of 17.60. The range size reflects the scope of influence of the variables. Thus, the SOC of neighboring soil samples exhibits strong spatial autocorrelation when the distance is less than 308.97 m. The nugget value can reflect the randomness of regionalized variables, whereas sill size represents the amplitude of variation for regionalized variables. The ratio of nugget variance ($C_0$, 21.67) to the total sill ($C_0 + C$, 39.27) was 55.18%. The ratio of nugget variance to the sill variance of variables was usually considered a norm to classify the spatial dependence of environmental factors [75]. Ratios of <25%, between 25% and 75%, and 75% indicate strong, moderate, and weak spatial dependence, respectively [76]. Thus, the moderate spatial dependence in this area indicated that extrinsic factors, such as sampling methods or other anthropogenic activities, weakened the spatial dependence caused by intrinsic factors, such as parent material and other geological characteristics [77]. The intrinsic factors strengthened the spatial dependence, whereas extrinsic ones accounted for weak spatial dependence [78].

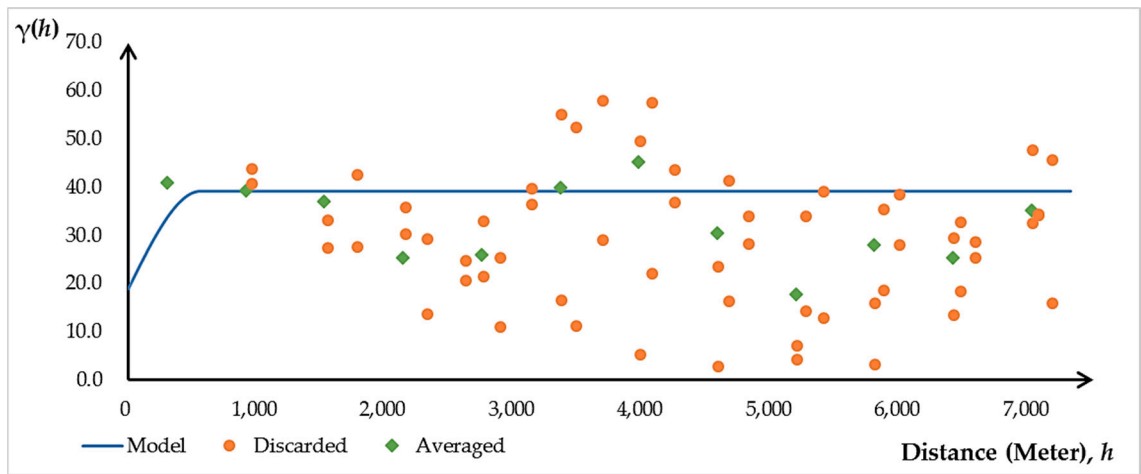

**Figure 9.** Experimental variogram and fitted models of SOC (soil organic carbon) through OK (ordinary kriging) interpolation.

The PLSR algorithm selects successive orthogonal factors that maximize the covariance between predictors (X, e.g., NDVI time series data) and response variables (Y, e.g., SOC). In this research, highly suitable auxiliary predictor data were extracted from the NDVI series data using PLSR. The suitable number of LVs was three, which was selected through cross-validation, to evaluate the SOC content. Table 4 shows the detailed information. The percentages of variance of the three LVs of NDVI were 93.69%, 1.52%, and 1.45%, and those of SOC were 55.48%, 10.64%, and 1.71%, respectively. The cumulative percent of variance of the three LVs (of NDVI) was 96.66%, indicating that these LVs

included 96.66% of the original NDVI information. Similarly, the cumulative LV of SOC was 67.87%, showing that this model explained 67.87% of the SOC information. This result indicated that the PLSR model can obtain substantial useful information for predicting SOC without redundant data.

**Table 4.** Predicted results of SOC via the PLSR model (percent variance captured by the regression model).

| | X-Block (NDVI) LVs | X-Block (NDVI) Cumulative | Y-Block (SOC) LVs | X-Block (NDVI) Cumulative |
|---|---|---|---|---|
| **1** | **93.69** | **93.69** | **55.48** | 55.48 |
| 2 | 1.52 | 95.22 | 10.68 | 66.16 |
| 3 | 1.45 | 96.67 | 1.71 | 67.87 |

Note: LVs: latent variables (or principal component); $RMSE_{CV}$: root mean square error of cross-validation; NDVI: normalized difference vegetation index; SOC: soil organic carbon.

In the SVM model, "epsilon ($\varepsilon$)-SVR" was used considering the runtime and efficiency in this study. In this model, the parameter cost was 3.16, and epsilon was 0.10, thereby presenting close fitting to the calibration data. Gamma was 0.032, which controlled the shape of the separating hyperplane. An increasing gamma usually increases the number of support vectors. The accuracy of the SVM model resulting from cross-validation was expressed by an $RMSE_{CV}$ of 4.69 and a determination coefficient of cross-validation ($R^2_{cv}$) of 0.33. Thus, the SVM method for regression was used to simulate the SOC content and demonstrated a strong capacity compared with the abovementioned methods. In the ANN method, the optimum number of nodes (i.e., four) in the hidden layer should be determined after comprehensive consideration of RMSE acquired by calibration and cross-validation to create the SOC prediction model based on cross-validation (Table 5). Evidently, this ANN model for regression can be effectively applied to predict soil properties with the dimensionality reduction of data and remove the data redundancy.

**Table 5.** Number of nodes in the hidden layer and corresponding precision in the ANN model.

| Number Nodes | RMSEcal SOC | RMSEcv SOC |
|---|---|---|
| 1 | 4.67 | 4.89 |
| 2 | 4.39 | 4.84 |
| 3 | 4.22 | 4.81 |
| 4 | 4.33 | 4.79 |

Notes: RMSEcal: root mean square error of calibration; RMSEcv: root mean square error of cross-validation; SOC: soil organic carbon.

### 3.4. Validation and Evaluation

Table 6 and Figure 10 show the validation indexes of SOC prediction by the five models. ANN exhibited the lowest RMSE of prediction ($RMSE_P$) of 3.718 and the highest coefficient of determination of prediction ($R^2_P$) of 0.391. This approach was the optimal method to predict SOC in the study area. The suboptimal prediction model was OK, with a slightly higher $R^2_P$ (0.372) and a lower $RMSE_P$ (3.727) than the other models, followed by SVM with $RMSE_P$ and $R^2_P$ of 3.753 and 0.361, respectively. Furthermore, PLSR demonstrated a better result than SLR with higher $R^2_P$ (0.283) but unexpectedly showed the highest $RMSE_P$ (4.087). The SLR model did not effectively perform with the lowest $R^2_P$ (0.281) and highest $RMSE_P$ (3.930). These results demonstrated that the NDVI time series data had high correlation with SOC and can be successfully used to predict SOC contents as good indicators of the primary ecological productivity of soil vegetation. 0The predicted SOC values were approximately close to the measured ones to some extent. Figure 10 illustrates that the predicted value obtained by the ANN method was close to the measured value (1:1 red line). The ANN model reflected high SOC values, whereas other methods suppressed high values and highlighted low values. The scatter plot showed that the predictive ability of the SVM method for high-value part was only

second to ANN. The scatter plots of PLSR and SLR methods were generally similar, whereas the OK model exaggerated the low value of SOC. On the whole, these findings indicated that ANN was the optimal model for SOC prediction. These findings were supported by Taghizadeh-Mehrjardi et al. [44] and Besalatpour et al. [79], who concluded that the ANN model shows the best performance for SOC prediction, followed by the SVM model. The proposed ANN models were more feasible than the other methods in extracting existing patterns among the ancillary data and SOC.

**Table 6.** Accuracy assessment of prediction models.

| | Modeling Accuracy | | Prediction Accuracy | |
|---|---|---|---|---|
| | RMSE | $R^2$ | RMSE | $R^2$ |
| Stepwise Linear Regression (SLR) | 4.863 | 0.270 | 3.930 | 0.281 |
| Ordinary Kriging (OK) | 3.549 | 0.524 | 3.727 | 0.372 |
| Partial Least Squares Regression (PLSR) | 4.970 | 0.230 | 4.087 | 0.283 |
| Support Vector Machine (SVM) | 4.269 | 0.453 | 3.753 | 0.361 |
| Artificial Neural Network (ANN) | 4.326 | 0.417 | 3.718 | 0.391 |

Notes: RMSE: root mean square error; $R^2$: coefficient of determination.

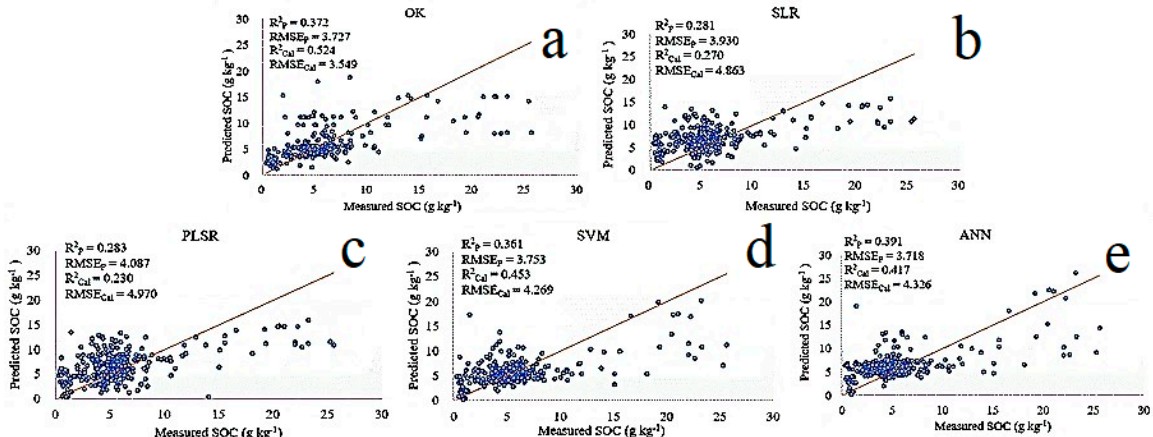

**Figure 10.** Scatter plot of measured and predicted SOC (soil organic carbon) values of the validation dataset by five models. Note: the red lines show 1:1 relationship and do not fit the data. (**a**): OK, (**b**): SLR, (**c**): PLSR, (**d**): SVM and (**e**): ANN.

### 3.5. Digital Mapping of SOC

The spatial outputs of digital SOC maps were mapped with extracted cultivated land via the aforementioned models (Figure 11). In general, the maps were relatively similar and showed strong spatial variation of SOC. High predicted SOC values occurred in the center, western, and northwestern parts of the study area, where the land was mainly covered by rice, lotus root, and oilseed rape (*Brassica napus* L.) with dense vegetation coverage. The high SOC contents in these parts of the research area could be explained by the long-term benign cultivation. Thereafter, low values were observed in the southern and eastern parts of the maps close to Yangtze River possibly because of the loss of soil nutrients in this area.

However, some evident differences were observed in the local details among the five methods. This finding was highly significant for the differences of SOC concentrations in the eastern part of this region. Figure 11d depicts that the predicted SOC content (0.5–3.5 g kg$^{-1}$) of the OK model was generally lower compared with the other maps with higher SOC values (3.5–10 g kg$^{-1}$) in the eastern part of the study area. As shown in the result of OK model (Figure 11d), the SOC content obtained by OK method was approximately low (the red part was small). However, from the perspective of cartographic quality, the OK model did not accurately show the local details of SOC content distribution

compared with other methods that produced many detailed patterns. This finding was because of the insufficient soil samples and the smoothing effect of OK interpolation. Thus, OK was unfit for predicting object values beyond the study region. The maps generated by the SLR and PLSR models are approximated in Figure 11a,e, which highlight the high values in the center of maps. Compared with the SLR method, the map of PLSR more strongly manifested low SOC values in the southern part of the study area. Moreover, the map obtained by SVM resembled that of the ANN model (0however, the map acquired by SVM tended more toward the middle and low values compared with that of the ANN model, which mostly presented middle and high values with lesser low values).

In summary, the SOC content from east to west of the study area showed the distribution ranging from low to high (0.3–40 g kg$^{-1}$). The finding verified that the spatial distribution of SOC content can be digitally mapped through the five models as expected. The NDVI time series data were closely related to the SOC content, which can be considered an indicator for spatial prediction.

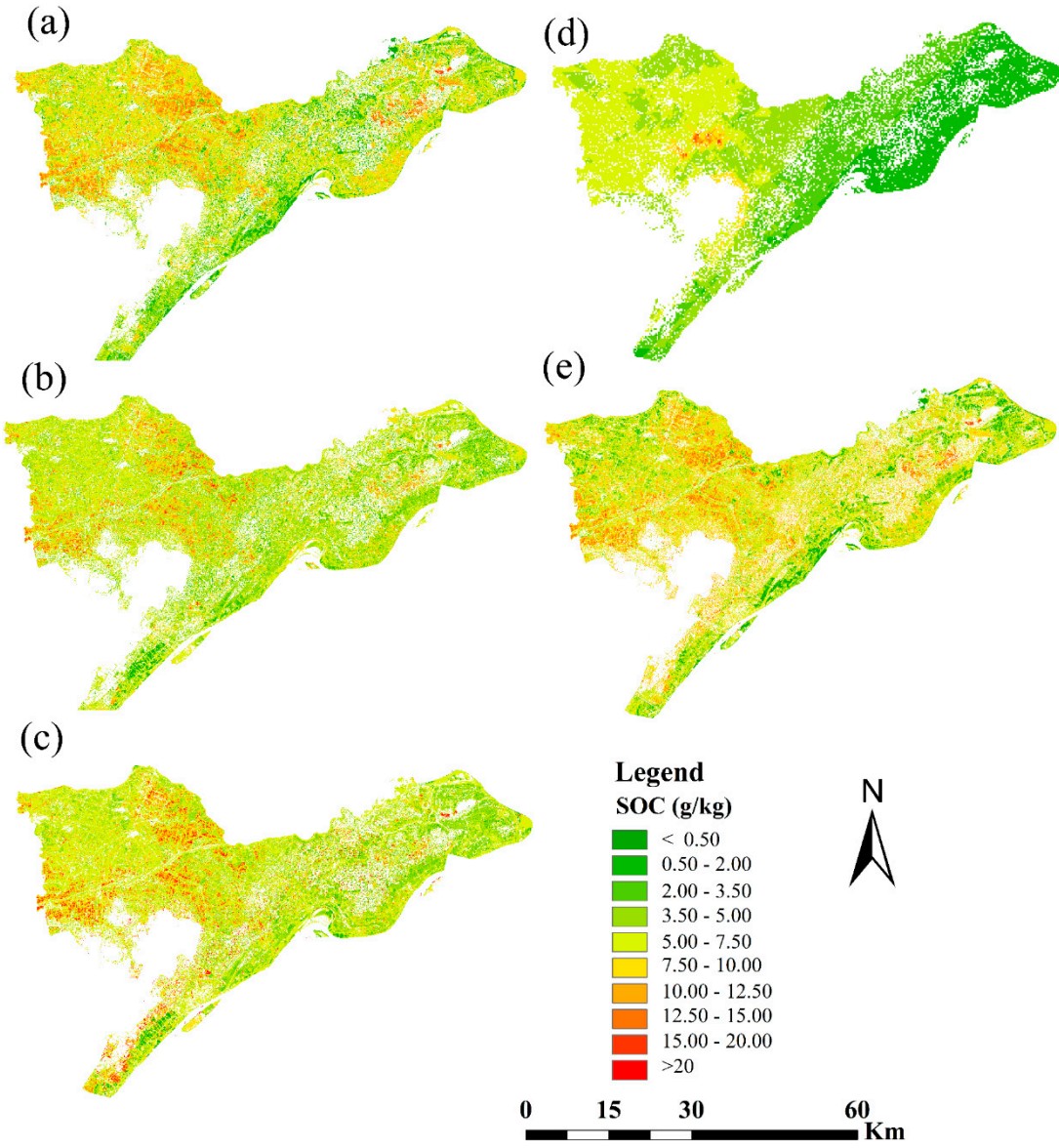

**Figure 11.** Spatial distribution of predicted SOC (soil organic carbon) content by (**a**) PLSR (Partial Least Squares Regression), (**b**) SVM (Support Vector Machine), (**c**) ANN (Artificial Neural Network), (**d**) OK (Ordinary Kriging), and (**e**) SLR (Stepwise Linear Regression).

## 4. Discussion

*4.1. Comparisons of Model Performance in SOC Prediction*

In this study, NDVI time series data were used to predict the SOC content in Honghu City via five approaches (SLR, OK, PLSR, SVM, and ANN). In summary, all these methods explained the spatial variance of SOC. Among these methods, ANN showed the optimal capacity to estimate the spatial distribution of SOC by providing sufficient useful information and decreasing redundant data. Similar conclusions were reported by Taghizadeh-Mehrjardi et al. [44], who mapped SOC lateral and vertical variances using data mining techniques (ANN, SVM, k-nearest neighbor, random forests, regression tree models, and genetic programming) and found that ANN is the best method to predict the SOC content. Meanwhile, the performance of SVM models to predict and map SOC was second to that of ANN. In reality, the SVM method was a feasible means for regressing and forecasting in agriculture, hydrology, meteorology, and environmental research [67–69]. Furthermore, Were et al. [66] confirmed the close performance of ANN and SVM models for spatially predicting and mapping the pattern of SOC stocks. The authors proposed that SVM and ANN methods should be calibrated, and the best result should be applied for the spatial prediction of target soil properties. Thus, the ANN model with the lowest $RMSE_P$ (3.718) and highest $R^2_P$ (0.391) was the best method for SOC prediction in this study. By contrast, the PLSR model performed slightly worse because it was based on traditional linear regression although the information of SOC was not. Kuang et al. [80] demonstrated that the ANN model performs better compared with PLSR, which may be attributed to a nonlinear behavior documented for SOC [81,82] and appeared to be overcome by the nature of ANN in solving nonlinear problems.

With regard to the OK model, some high values were underestimated, and some lower values were overestimated because of the filtration of partial information and reservation of the whole tendency [83]. In particular, OK exhibited a smoothing effect, which removed the maximum and minimum values of the original data. Meanwhile, the OK model mapped the spatial distribution of SOC in the study region with the soil samples and cannot be extended to neighboring regions. Thus, the application of the OK method was limited [84]. Gu [85] summarized that OK interpolation is mainly used in field scales or some large areas with relatively consistent management measures. However, the prediction effect of OK interpolation on the spatial differentiation of SOC is unsatisfactory in some areas with complex terrain and land use. The SLR model performed the worst, with the lowest $R^2_P$ (0.281) and slightly higher $RMSE_P$ (3.930). This finding can be explained by the collinearity among the predictors. The existence of multicollinearity between environmental predictors might be inevitable because soil formation is determined by the interaction of various soil-forming factors, leading to the weak stability and poor prediction ability of the established soil attribute prediction model [64]. In the SLR model, the collinearity problem was evaluated through multicollinearity diagnosis and computation of variance inflation factor (VIF), as shown in Tables 7 and 8. In principle, VIF > 10 indicates a collinearity problem [24]. Although the VIF values showed no evident collinearity (Table 1), the lowest eigenvalue was 0.01, and the highest condition index was 27.10, indicating a slight collinearity problem with SLR, as shown in in Table 7. Thus, the stability of simple multiple linear regression model was poor. These conclusions were supported by Wang et al. [64], who summarized the underperforming result of SLR models.

**Table 7.** Collinearity diagnosis of environmental factors in the SLR model.

| Dim | Eigenvalue | Condition Index | Variance Proportions | | | | | | | |
|---|---|---|---|---|---|---|---|---|---|---|
| | | | Con | $NDVI_{Feb.}$ | $NDVI_{Jul.}$ | $NDVI_{Sept.}$ | $NDVI_{Jan.}$ | $NDVI_{Aug.}$ | $NDVI_{Jun.}$ | $NDVI_{Apr.}$ |
| **1** | **7.52** | 1.00 | 0.00 | 0.00 | 0.00 | 0.00 | 0.00 | 0.00 | 0.00 | 0.00 |
| 2 | 0.20 | 6.09 | 0.00 | 0.07 | 0.00 | 0.03 | 0.09 | 0.00 | 0.10 | 0.00 |
| 3 | 0.08 | 9.44 | 0.00 | 0.01 | 0.00 | 0.33 | 0.04 | 0.02 | 0.38 | 0.00 |
| 4 | 0.08 | 9.61 | 0.00 | 0.01 | 0.02 | 0.05 | 0.01 | 0.18 | 0.04 | 0.24 |
| 5 | 0.05 | 12.92 | 0.02 | 0.56 | 0.00 | 0.02 | 0.06 | 0.05 | 0.08 | 0.43 |
| 6 | 0.03 | 15.69 | 0.38 | 0.01 | 0.02 | 0.27 | 0.41 | 0.00 | 0.25 | 0.12 |
| 7 | 0.03 | 16.35 | 0.30 | 0.34 | 0.00 | 0.18 | 0.36 | 0.17 | 0.06 | 0.20 |
| 8 | 0.01 | 27.10 | 0.30 | 0.00 | 0.95 | 0.12 | 0.03 | 0.58 | 0.09 | 0.01 |

Note: Con is the constant of function, $NDVI_{Feb.}$ is the NDVI (normalized difference vegetation index) data in February, and $NDVI_{Jul.}$ is NDVI data in July, etc.

**Table 8.** VIFs for the SLR model.

| | $NDVI_{Feb.}$ | $NDVI_{Jul.}$ | $NDVI_{Sept.}$ | $NDVI_{Jan.}$ | $NDVI_{Aug.}$ | $NDVI_{Jun.}$ | $NDVI_{Apr.}$ |
|---|---|---|---|---|---|---|---|
| Tolerance | 0.48 | 0.37 | 0.63 | 0.39 | 0.41 | 0.74 | 0.60 |
| VIF | 2.08 | 2.71 | 1.59 | 2.57 | 2.46 | 1.35 | 1.66 |

Note: VIF is the variance inflation factor, $NDVI_{Feb.}$ is the NDVI (normalized difference vegetation index) data in February, and $NDVI_{Jul.}$ is the NDVI data in July, etc.

In summary, all the results showed a strong correlation between NDVI time series data and SOC. In particular, NDVI time series showed interpretability for SOC. However, the $R^2$ values of all the models in this study were low. This condition was because the SOC content was related to NDVI and influenced by soil type and structure, precipitation, climate, and other factors because of the diversity of soil-forming factors. Additional pivotal factors combined with the time series method should be calibrated for the spatial prediction of soil properties. Meanwhile, time periods of environmental factors should be considered to apply the highly appropriate time series. For cartographic quality, machine learning and traditional regression methods based on pixel-raster cartographic methods could obtain relatively fine maps and show the interior details of SOC spatial distribution in Honghu City. Evidently, the spatial distribution of SOC mapped by the OK model had large and rough raster polygon because of its smoothing effect. Thus, the digital SOC map applied by the ANN model was considered the optimal one comprehensively considering the prediction accuracy and mapping requirements.

*4.2. Superiority of Time Series NDVI Approach*

This study has two main significances, namely, to discuss whether the NDVI time series can play an important role when the terrain factors in plain areas cannot provide a huge influence on the spatial variance of the SOC content and to provide a method to indirectly evaluate SOC via crop growth when the soil of earth surface is covered by vegetation.

4.2.1. Exploring the Deep Mechanism of the Relationship Between SOC and NDVI Time Series

SOC is a major indicator of soil fertility and plays an important role in ecosystem productivity, agricultural ecosystem function, and farmland fertility [86]. Thus, the SOC content affects crop yield to a large extent. NDVI is strongly correlated to vegetation GPP [87]. GPP is derived from the calculation of photosynthetic effective radiation (PAR), namely, the famous light energy utilization equation [87–89]:

$$GPP = fAPAR \times PAR \times LUE, \tag{7}$$

$$APAR = fAPAR \times PAR, \tag{8}$$

where fAPAR is the fraction of absorbed photosynthetically active radiation, APAR is the absorbed photosynthetically active radiation, PAR is the photosynthetically active radiation, and LUE is the

vegetation light energy utilization. Several studies have shown that fAPAR and LUE are strongly correlated with VI [90–93]. Thus, the following formula can be derived [94]:

$$GPP \propto PAR \times VI, \tag{9}$$

A feedback loop occurs where production feeds carbon into the soil, and high SOC improves the soil, indicating large production. This result was supported by Heinemeyer et al. [95], who considered GPP implicate the turnover of SOC to some extent. Kimball et al. [96] used GPP as an auxiliary to estimate surface (<10 cm depth) SOC stocks. Dong et al. [90] used NDVI to estimate fAPAR. Chang et al. [97] used NDVI and ratio vegetation index (RVI) to calculate fAPAR. Crop growth in cultivated lands was influenced by soil, which was reflected in remote sensing images. Thus, time series NDVI can potentially conduct soil mapping in plain areas. On this basis, NDVI can predict SOC stocks.

Considering only the environmental factors at a certain time is inaccurate because of the complexity of soil formation. Thus, the time series characteristics of auxiliary variables should be considered to effectively reflect the relationship between environmental factors and target variables. Meanwhile, as previously mentioned, the sum of NDVI is correlated to GPP (the sum of APAR radiation, which is NDVI times incident PAR). Single-date NDVI is not correlated to SOC and production. Gitelson et al. [93] used multitime NDVI to estimate fAPAR. Thus, the NDVI time series data in this study, which consider a vital factor (i.e., time), can closely indicate their correlation. For the effect of regression using single-data NDVI, Table 9 shows the unsatisfactory results (with all $R^2$ <0.1). This finding indicated that single-data NDVI cannot be used to predict SOC.

**Table 9.** Single-data NDVI regression analysis.

|  | Modeling Accuracy | | Prediction Accuracy | |
| --- | --- | --- | --- | --- |
|  | **RMSE** | $R^2$ | **RMSE** | $R^2$ |
| $NDVI_{Jan.}$ | 5.542 | 0.031 | 4.693 | 0.020 |
| $NDVI_{Feb.}$ | 5.407 | 0.077 | 4.710 | 0.032 |
| $NDVI_{Mar.}$ | 5.498 | 0.046 | 4.688 | 0.023 |
| $NDVI_{Apr.}$ | 5.441 | 0.065 | 4.793 | 0.058 |
| $NDVI_{MAy}$ | 5.608 | 0.007 | 4.747 | 0.004 |
| $NDVI_{Jun.}$ | 5.529 | 0.035 | 4.574 | 0.064 |
| $NDVI_{Jul.}$ | 5.536 | 0.032 | 4.625 | 0.060 |
| $NDVI_{Aug.}$ | 5.568 | 0.021 | 4.600 | 0.071 |
| $NDVI_{Sept.}$ | 5.488 | 0.049 | 4.675 | 0.041 |
| $NDVI_{Oct.}$ | 5.630 | 0.001 | 4.754 | 0.001 |
| $NDVI_{Nov.}$ | 5.565 | 0.022 | 4.799 | 0.001 |
| $NDVI_{Dec.}$ | 5.583 | 0.016 | 4.692 | 0.021 |

The predictive effect of short time series was evaluated. Two short NDVI time series data, which were correlated to summer and winter crop production, respectively, were applied to model ANN and SVM. The results (Table 10) showed that the NDVI time series data of summer crop production exhibited better precision ($R^2_P$ of ANN 0.281, $RMSE_P$ of ANN 4.119, $R^2_P$ of SVM 0.343, $RMSE_P$ of SVM 3.863) than that of winter crop production ($R^2_P$ of ANN 0.173, $RMSE_P$ of ANN 4.980, $R^2_P$ of SVM 0.162, $RMSE_P$ of SVM 4.908). This finding manifested that short NDVI time series (related to crop production) can be used for SOC prediction to some extent, but the prediction accuracy of short time series was lower compared with long time series, which was particularly evident for winter crop production. This condition was because Honghu City had more summer crops that winter crops (the most yield crop were lotus root and rice, which were summer crops). Another reason was because short time series NDVIs of summer were true time series (from 2013.4 to 2013.9), whereas that of winter was not. Although the entire NDVI time series under multiple years can be used to predict SOC regarded as pseudo time series, the short time series was susceptible to be influenced considerably. At the same

time, the accuracy of these short time series predictors for the SVM model was better than that of the ANN model. This condition discussed the performance of SVM and ANN methods, as suggested by Were et al. [66]. The SVM method obtained better results ($R^2_P$ for summer 0.343, $RMSE_P$ for summer 3.863, $R^2_P$ for winter 0.162, $RMSE_P$ for winter 4.908) for prediction using short NDVI time series data than that of ANN ($R^2_P$ for summer 0.281, $RMSE_P$ for summer 4.119, $R^2_P$ for winter 0.137, $RMSE_P$ for winter 4.980), and the entire NDVI time series data provided close performance of ANN ($R^2_P$ 0.391, $RMSE_P$ 3.718) and SVM ($R^2_P$ 0.361, $RMSE_P$ 3.753). Thus, the SVM method was selected for SOC prediction using short time series.

**Table 10.** Two short subtime series predictors for model analysis.

| Predictors | Modeling Accuracy | | Prediction Accuracy | |
|---|---|---|---|---|
| | RMSE | $R^2$ | RMSE | $R^2$ |
| Subset1(summer) for ANN: $NDVI_{Apr.}$ $NDVI_{May}$ $NDVI_{Jun.}$ $NDVI_{Jul.}$ $NDVI_{Aug.}$ $NDVI_{Sept.}$ | 4.818 | 0.269 | 4.119 | 0.281 |
| Subset2 (winter) for ANN: $NDVI_{Jan.}$ $NDVI_{Feb.}$ $NDVI_{Mar.}$ $NDVI_{Oct.}$ $NDVI_{Nov.}$ $NDVI_{Dec.}$ | 4.855 | 0.141 | 4.980 | 0.137 |
| Subset1 (summer) for SVM: $NDVI_{Apr.}$ $NDVI_{May}$ $NDVI_{Jun.}$ $NDVI_{Jul.}$ $NDVI_{Aug.}$ $NDVI_{Sept.}$ | 4.158 | 0.477 | 3.863 | 0.343 |
| Subset2 (winter) for SVM: $NDVI_{Jan.}$ $NDVI_{Feb.}$ $NDVI_{Mar.}$ $NDVI_{Oct.}$ $NDVI_{Nov.}$ $NDVI_{Dec.}$ | 4.575 | 0.237 | 4.908 | 0.162 |

### 4.2.2. Effect of Terrain Factors on SOC Prediction

Digital mapping has become a crucial and popular method to investigate the spatial distribution of soil properties, which provide substantial details. Presently, numerous studies have focused on exploring a feasible and reliable method to acquire spatial SOC information [24,53,98]. However, all these methods have considered topography as a necessary influential factor. In general, terrain attributes are pivotal factors that affect the SOC stock in areas with complex and varied topography, such as mountainous regions [98,99]. However, in plains or small-scaled areas, the spatial distribution of terrain is flat and has indistinctive variation. As previously mentioned, the correlation between topographic parameters (surface roughness, DEM, slope, and TPI) and SOC was low (0.014, 0.129, 0.021, and 0.044). The four terrain variables were used as predictors to model ANN and SVM to prove that topographic factors had small effect on SOC prediction in this plain area. However, the best modeling methods failed to produce good results (Table 11). Hence, the effect of topography for SOC was negligible in this small-scale plain area. Similar conclusions were reported by Song et al. [100], who regarded that local terrain attributes played less important role than other predictors at a small scale. Johnson et al. [101] concluded that terrain attributes can capture large-scale influences of soil transport but not those occurring at a specific point. Thus, the overuse of terrain factors in small-scale plain areas reduces the prediction accuracy and increases the calculation complexity.

**Table 11.** SOC with respect to topographic variation (digital elevation, slope, surface roughness, and topographic position index (TPI)) for ANN and SVM models.

| | Modeling Accuracy | | Prediction Accuracy | |
|---|---|---|---|---|
| | RMSE | $R^2$ | RMSE | $R^2$ |
| ANN | 5.580 | 0.019 | 4.829 | 0.027 |
| SVM | 5.199 | 0.021 | 5.305 | 0.012 |

Notes: ANN is artificial neural network, SVM is support vector machine.

In summary, in plains and gentle undulating terrains, easy-to-measure environmental factors, such as topography (elevation and slope), do not co-vary with soils that have spatial variations that are difficult to effectively reflect [29,102]. Zhu et al. [29] evaluated the change patterns (dynamic

feedback patterns) of the land surface, such as those captured daily by remote sensing images during a short period (6–7 d) after a major rain event, which can be used to differentiate soil types. However, the main problem is that remote sensing images require a short round trip period of the satellite. Meanwhile, data collection is susceptible to cloud, snow, and rain. In particular, remote sensing images require high temporal and spatial resolutions, which cause immense difficulty to high-precision digital soil mapping.

### 4.3. Limitations of Research

In this study, time series NDVI data were used as predictors to evaluate SOC content and map its spatial distribution. Overall, the results were relatively satisfactory but can still be improved. First, many reasonable sampling methods, such as sampling location and number of samples settings, were recommended. This process ensures that the measured data are reliable and close to real value. Second, for the time series approach, continuous and high-quality images of true time series may lead to better accuracy. Moreover, the relationship between VI and soil based on time series characteristics should be determined to improve the mapping accuracy. Future studies should combine the time series characteristics of multiple important auxiliary variables to predict the target soil properties given the complexity of environmental factors.

## 5. Conclusions

Traditional environmental factors, such as terrain, landscapes, climatic, and vegetation types, exhibit similar spatial distribution characteristics in plain or flat areas. These factors cannot effectively reflect the spatial variation of soil properties. Thus, NDVI time series data were used as the auxiliary variables for digital soil mapping using five methods, namely, OK, SLR, PLSR, SVM, and ANN.

(1) The results demonstrated that NDVI time series was correlated with SOC stock. A significant positive correlation was observed between the SOC content and NDVI in July and August. By contrast, a significant negative correlation was observed between the SOC content and NDVI in January, February, March, April, November, and December. This finding was attributed to the cultivation of farming work and the phenophase of the crops that influenced the land surface vegetation landscapes.

(2) The comparison result of different methods showed that ANN was the overall best method with the lowest $RMSE_P$ of 3.718 and highest $R^2_P$ of 0.391, followed by SVM ($RMSE_P$ = 3.753, $R^2_P$ = 0.361), OK ($RMSE_P$ = 3.727, $R^2_P$ of 0.372), PLSR ($RMSE_P$ = 4.087, $R^2_P$ = 0.283), and SLR ($RMSE_P$ = 3.930, $R^2_P$ = 0.281). Thus, ANN was the optimal model to predict SOC using the NDVI time series.

(3) The SOC maps estimated by the five models were similar. The SOC content from east to west of the study area showed distribution ranging from low to high (0.3–40 g kg$^{-1}$). However, the local details clearly indicated that OK interpolation smoothed the result, and the maps generated by the SLR and PLSR models highlighted high values in the center of the maps. Moreover, the map acquired by SVM tended more toward the middle and low values compared with that of the ANN model, which mostly presented middle and high values with lesser low values. These results confirmed that the spatial distribution of the SOC content can be digitally mapped through NDVI time series.

(4) The prediction of SOC using single-data NDVI showed unsatisfactory accuracy, indicating the unpredictability of single-data NDVI compared with multi-time NDVI. The prediction results of two short NDVI time series, which were correlated to summer and winter crop production, respectively, manifested that short NDVI time series can be used for SOC prediction to some extent, but its prediction accuracy was lower than that of long time series. In addition, the correlation between topographic parameters and SOC was low. The terrain variables used as predictors in the model failed to produce good results. Hence, the effect of topography for SOC was negligible in this small-scale plain area.

Finally, this study provided an example for efficiently mapping regional SOC to overcome the selection of auxiliary variables in plains. The NDVI time series data exhibited huge potential in predicting SOC, and ANN extracted substantial useful information for digital soil mapping. Thus, the approach that combines the time series characteristics of multiple important auxiliary variables via ANN should be investigated and used in other areas for other soil properties. An enhanced understanding of the spatial distribution of soil properties facilitates soil management and accelerates agricultural production and ecological planning.

**Author Contributions:** Conceptualization, G.L.; Data curation, Y.Z.; Formal analysis, Y.Z.; Funding acquisition, H.Z.; Investigation, M.L.; Methodology, G.L.; Project administration, G.L.; Resources, Y.C.; Software, S.W.; Supervision, Q.J.; Validation, T.S.; Writing—original draft, Y.Z.; Writing—review & editing, G.L.

**Funding:** This research was funded by the National Natural Science Foundation of China, grant number (41471179, 41371227); the Basic Research Program of Shenzhen Science and Technology Innovation Committee (No. JCYJ20170302144323219); the Natural Science Foundation of Hubei, grant number (2018CFB372); the National Key R&D Program of China (No. 2017YFC0506200); and the National Undergraduate Innovation and Entrepreneurship Training Program, grant number (201810504023, 201810504030).

**Conflicts of Interest:** The authors declare no conflict of interest.

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
