# Peer review of "Prediction of Soil Organic Carbon based on Landsat 8 Monthly NDVI Data for the Jianghan Plain in Hubei Province, China"

_remotesensing, doi:10.3390/rs11141683_

Round 1
Reviewer 1 Report
The theoretical foundation of the study is much better because of the explicit link between plant productivity and SOC. Furthermore, various alternative hypotheses are better explored. Comparing remote sensing with SOC has been undertaken many times over the years, but geospatial SOC maps are very important for understanding the earth system and potential sequestration of atmospheric CO2 as soil organic matter. Thus, this subject should be revisited when new techniques are available.
Major comments:
1. In the newly-added text, the authors make many basic errors about the carbon cycle, so I wonder how many other statements in subjects I know little about are also in error. Whereas it is much easy to reject the manuscript for this reason, the authors made a reasonable effort to address my previous concerns.
2. The weak correlation between SOC and topography and other variables may be the result of cultivation or other soil disturbances. However, the process-based variables affecting SOC should be tested before searching for other correlations such as a time series of NDVI.
3. A mistake which is frequently seen (unfortunately) in published studies is where two or more populations are combined which creates a significant result. Here there is potential that two populations (different weather), each with no relationship between time-series NDVI and SOC, are combined to create a dataset that has a significant relationship. The goal of the short time series (L 667-690) was to determine that the pseudo times series created the best SOC map simply by chance. With fewer points, accuracy will decrease with any of the methods used in this study. However, if the ANN results are valid, then the short true time series should be similar to the pseudo time series. This was also the purpose for Figure 2, which shows the pseudo time series does not vary considerably from the true time series.
Specific comments:
L 32: delete “innovatively” because this type of study has been done several times before
L 40: NDVI time series is a method, not a value
L 45-46: “OK” is a subjective assessment. The difference between two correlation coefficients may be tested for significance using Fisher’s r-to-z transformation.
L 68: “SOC content is based on laboratory analysis of field samples.
L 72: delete sentence starting “Quantifying”
L 78-81: Do you have references for this statement? Any statistical method is limited by the number of samples and outliers.
L 84: “over a long time.”
L 90: delete “multi-factors”
L 91: delete “of SOC”
L 92-94: Replace 2 sentences with, “Large variation in topography creates large variations in climate and other environmental variables related to SOC, leading to strong statistical relationships. However, in areas of little topography, such as plains, environmental variations are small making it harder to develop accurate predictions of SOC.
L 105-106: a model derives remote sensing images? Please reword.
L 111-118: delete and start sentence with “Changes”
Note, references 31 and 32 are about agriculture being the largest sources of methane and nitrous oxide, two very important greenhouse gases, but these gases are not CO2. There are large fluxes of CO2 from soils by plant respiration and SOM decomposition, but these fluxes are balanced by photosynthesis.
L 121: replace “remains” with “residue”
L 229: backwards, weather is the independent variable, crop production is the dependent variable
L 230: “little variation”
L 231: delete “is close”
L 231: delete next sentence
L 241, 251, 268, 285, 315: spell out abbreviation in heading
Figure 3: Gene X and Gene Y?? Need to create your own schematic. Need to be consistent on use of SVR and SVM.
L 298-306: What are the parameter names? The Greek letters ε and ν are symbols that represent some parameter, just like λ in Eq. 3 represents value weights. I did not intend for you to use the letter names as the parameter names. You have definitions of ε and ν in lines 307-314. Therefore the two paragraphs need to be rewritten to logically introduce SVM.
L 323-325: ANN’s are “black boxes” because the reasons for a specific result are unknown. Frequently, wavelengths selected for hyperspectral sensors have no relationship with spectral features known to be related to SOC. For scientific research, my preference is to get the wrong SOC, for which I can figure out what went wrong, instead of getting the correct answer for unknown reasons. However, ANN’s are useful when the objective is to get a good map.
Reviewer 2 Report
General comments:
The authors tried their efforts in revising the manuscript, and it was improved compared to the previous version. However, my concern is that the relationship between the predicted SOC and measured SOC is not good enough. I suggest the authors add the accuracy of different method especially the accuracy of SOC predicted by remote sensing images.
The resolution of Landsat-8 is 30m by 30m, but only one sample point in one Landsat pixel according to your sample strategy? The bias caused by different scales of remote sensing data and measured data should be considered in the soil sampling. What did you do to reduce this bias?
Detailed comments:
L79-81: revise this sentence.
L85: “Wang, et al.” -> ”Wang et al.” same for others
L92: “These demonstrates…”-> “These demonstrate…”
L217: There are many supervised classification methods in ENVI: Parallelepiped, minimum distance, maximum likelihood,..., which one did you use?
L220: Delete “ENVI classic tool”.
L227: The text in Figure 2 is not clear. Make the resolution higher.
L238: “3δ criteria” -> add reference
L744: “Strong correlation” ?
Please revise the style of reference
Author Response
Please see the attachment

This manuscript is a resubmission of an earlier submission. The following is a list of the peer review reports and author responses from that submission.
Round 1
Reviewer 1 Report
General comments:
This paper compared the predictions of SOC using NDVI time series based on five different models. It indicates that NDVI time series has the potential to map SOC in plain areas. The English language and style are good. However, as there are a lot of factors that can affect crop growth (or NDVI), the correlation is weak. It might be worthwhile to improve the discussion section and discuss the limitations of this method.
Detailed read-through comments:
L33: R2P-> R2P ?
L51: SOC is meaningful …-> SOC is a meaningful …
L137: km2->km2
L152-L153: delete “Location of the study area and spatial distribution of the calibration (407) and validation (271) datasets. Figure 1.”
L153-L154: “Soil samples from each location were composites of five soil samples taken from the four corners and the center of a 1 m × 1 m square area”-> “For each location, five soil samples were taken from the four corners and the center of a 1 m × 1 m square area.
L167-L169: Does the SOC vary with years? Please explain here why the image from different years can be used. Did you mask the clouds when there are clouds? If not, how do you assure the sample sites are not covered by clouds?
L170-L171: You can download the surface reflectance data directly.
L173: What kind of supervised classification? What is the accuracy of the classification result?
L183: “abnormal values were rejected”->What is the rule to determine abnormal values?
L230: SVR model-> Full name when mentioning it for the first time. It is a little bit confusing between SVR and SVM before you give the full name of SVR. You should explain it earlier.
L325: delete “Error! Reference source not found.”
L331: Harvest time was in March or May? You mentioned in L334 that the harvest time for wheat is May.
L342: is the SOC content in-situ measured SOC?
L343: Presented in Figure 6?
L347-L348: Why other vegetation may cause negative correlation?
L360: Did you mention how to remove the multicollinearity somewhere?
L360-L361: These data are only used for SLR model or for all the models?
L431: Table 6 and ?
L567: “round-trip period of the satellite,”-> “round-trip period of the satellite.”
L574: The growth status of vegetation are affected by a lot of factors, such as Nitrogen, soil moisture, and SOC etc. What is the relationship between these factors? What is the contribution of SOC to vegetation?
L575-L585: This part better goes to the Introduction section. Contents related to the results should be discussed here (e.g. the accuracy comparison between using single NDVI and NDVI time series, and reasons). In addition, the limitations of this method should also be discussed.
Do the crop types change each year?
Since the exposure of soil surface will lead to inaccuracy by using NDVI information, is it possible to use a threshold to determine whether the image will be used or not?
Reviewer 2 Report
The titled of the manuscript is very promising. However, the method, results and discussion is very poor. The authors get some NDVI time series and correlate to Soil Organic Carbon (SOC) using different mathematical approach. In case we are living in the 90’s, the manuscript would be considered very interesting and high novelty. Nevertheless, there area thousands of papers about SOC retrievals using NDVI and statistical models. I did not see any new approach in this paper in comparison to those mentioned on the references. I have to recommend a rejection.
Just in case, Figure 7 is horrific.
Reviewer 3 Report
Major comments:
1. On line 68, other studies are merely correlations (true). But isn’t this study also a mere correlation? Perhaps. An important point about NDVI time series is the sum of NDVI is correlated to vegetation gross primary production (actually, it is the sum of absorbed photosynthetically active radiation, which is NDVI times incident PAR). So you have the possibility of a mechanism as to why NDVI time series predicts SOC. There is a feedback loop where production feeds carbon into the soil, and high SOC improves the soil so production is higher. This also explains why single-date NDVI is not well correlated to SOC, because single-date NDVI is not well correlated to production. Adding a paragraph to the discussion should be enough to make this point.
2. The monthly NDVI data are not a true time series because the data were acquired over multiple years, so vegetation is not affected by the same sequence of monthly weather (precipitation in one month does not affect plant production in the next month). A new figure is necessary, comparing the long-term average monthly weather (total precipitation, monthly-average high air temperature and monthly-average low air temperature) with the monthly data for each date in Table 1. Monthly high and low air temperatures are more related to crop production than is average daily air temperature. If the monthly data are close to the long-term averages, the data in Table 1 could be considered an artificial or pseudo time series.
3. Following up on comment 1, this study will be much more significant if the monthly NDVI data are causally related to one or more factors of soil formation. Comments 3a and 3b below may be considered as control analyses to eliminate the possibility that the correlations are spurious.
3a. From Table 1, there are two smaller time series: summer 2014 (4-9) and winter 2015-16 (10-12, 2), which should be more highly correlated to summer and winter crop production, respectively. The two short actual time series should be put through the same analyses. If this work amounts to nothing then this effort is only worth a sentence in the results.
3b. Lines 518-519: is this a sentence from Gu [64], a summation of several other studies, or a result from this study? Using only the Hubei plain was a good idea because the terrain is not complex, but there is still topographic variation. Surface and ground water transport decrease production in the higher areas and increase production in adjacent low areas. Vegetation is generally more productive in the lower areas leading to the feedback loop described in comment 1 above. Therefore in addition to ordinary kriging, co-kriging SOC with respect to topographic variation (digital elevation minus the overall trend in elevation) would serve as a control analysis. Like in comment 3a, if this work amounts to nothing then it is only worth a short paragraph in the methods (co-kriging) and a sentence in the results. However, I think it is likely that the results of co-kriging will be more significant than the ordinary kriging.
Minor comments:
Title: “Prediction of Soil Organic Carbon based on Landsat-8 monthly NDVI data for the Jianghan Plain in Hubei Province, China”
L 91-95: references needed
Table 1: what is the “Others” column? With no information, it should be deleted.
L 242-257: I would prefer the italicized lower-case Greek letters instead of epsilon and nu spelled out. Please treat this as another abbreviation, spell out first with Greek letters in parentheses.
L 294: Equation 5?
L 325: error in text
L 330: Species binomials are in Latin, and need to be italicized. However, don’t italicize the “L”.
Figure 8. Mention in caption that the lines show the 1:1 relationship, and are not a fit the data.
L 563-583: Four or more authors need to be reduced to et al. (Zhu et al. [26], Shen et al. [31], and Song et al. [22]). Three authors may be either spelled out or reduced to et al., depending on journal style. Because the first author’s name is hyphenated, probably et al. is preferred (Taghizadeh-Mehrjardi et al. [30]).
Reviewer 4 Report
I will describe some doubts below:
A) Lines 182 - 186
Among the 678 soil sampling points, are there samples in clouded areas? Because the OLI 2013-12-06 image presented 49.93% cloud coverage.
If there are samples collected in clouded areas, what is the influence on the time series for this loss of NDVI quality?
B) Line 230
Need to describe the acronym SVR
SVR - Support Vector Regression
C) Please check the text:
Lines 325 - "Error! Reference source not found"
Line 431 - "Table 6 and 0show the...."
Line 440 - "As presented in 0the predicted ..."
Line 469 - "are approximated in 0a and 9e, which ...."
Line 472 - "ANN model (0However ..."
Lines 486 and 488 - " Authors should discuss the results and how they can be interpreted in perspective of previous studies and of the working hypotheses. The findings and their implications should be discussed in the broadest context possible. Future research directions may also be highlighted
Line 526 - "andTable 8"
D) Necessary more information about:
Lines 342 and 343 - "The correlation analysis results of SOC content and NDVI by Pearson correlation coefficient (r) are presented in 0".
I did not understand the number 0 in the context of the phrase.
Line 593 - The maximum Pearson correlation coefficient between SOC (soil organic carbon) and NDVI igual |0.30|, can be considered as strong correlation?
Figure 6 shows the monthly correlation, how to explain the conclusion of strong correlation for the entire time series?